# “Girls Aren’t Meant to Exercise”: Perceived Influences on Physical Activity among Adolescent Girls—The HERizon Project

**DOI:** 10.3390/children8010031

**Published:** 2021-01-07

**Authors:** Emma S. Cowley, Paula M. Watson, Lawrence Foweather, Sarahjane Belton, Andrew Thompson, Dick Thijssen, Anton J. M. Wagenmakers

**Affiliations:** 1Research Institute for Sport and Exercise Sciences, Room 1.22 Tom Reilly Building, Byrom Street Campus, Liverpool John Moores University, Liverpool L3 5AF, UK; e.s.cowley@ljmu.ac.uk (E.S.C.); p.m.watson@ljmu.ac.uk (P.M.W.); l.foweather@ljmu.ac.uk (L.F.); D.Thijssen@ljmu.ac.uk (D.T.); 2Childhood and Physical Activity Research Cluster, School of Health and Human Performance, Dublin City University, D09 Y5NO Dublin, Ireland; sarahjane.belton@dcu.ie; 3Wolfson Centre for Personalised Medicine, Institute of Systems, Molecular and Integrative Biology, University of Liverpool, Liverpool L69 3BX, UK; andrew.thompson@liverpool.ac.uk; 4Radboud Institute for Health Sciences, Department of Physiology, Radboud University Medical Centre, 6525 GA Nijmegen, The Netherlands

**Keywords:** physical activity, adolescence, focus groups, girls, socioecological model, qualitative, exercise

## Abstract

Background. Adolescent girls are less active than boys, with approximately 10% of girls in Ireland and the United Kingdom meeting the minimum recommended daily physical activity (PA) guidelines. This study investigated factors perceived to influence PA among adolescent girls from low socioeconomic areas in order to inform the design of a future intervention (The HERizon Project). Methods. A total of 48 adolescent girls (13–18 years) from low socioeconomic areas of the United Kingdom and Ireland participated in focus groups (*n* = 8), to explore perspectives of physical activity and the influence of gender within this. Focus groups were thematically analyzed and interpreted within a socioecological framework. Results. Most girls enjoyed PA and were aware of its benefits. They identified both barriers and facilitators to PA at intrapersonal (fear of judgement and changing priorities WITH age), interpersonal (changing social pressures and support from others) and organizational (delivery of PE) levels. Gender inequality was a multilevel factor, crossing all socioecological levels. Conclusion. Although many adolescent girls enjoy PA, their experiences appear to be limited by a fear of judgement and an overarching sense of gender inequality. Future interventions, such as the HERizon Project, should address influences at intrapersonal, interpersonal and organizational levels to promote positive PA experiences for adolescent girls.

## 1. Introduction

Regular physical activity (PA) is associated with numerous physical and psychological health benefits for adolescents, including improved cardiometabolic health [1], healthy weight management [2], cognitive function [3], psychosocial skills [4] and mental wellbeing [5]. A survey conducted with 1.6 million participants found that less than 15% of the global adolescent population are meeting the recommended PA guidelines of at least 60 min moderate to vigorous PA (MVPA) per day across the week [6]. This report shows a gender disparity as adolescent females are less active than males, particularly females living in low socioeconomic areas [6]. This is of concern as the rate of obesity and type II diabetes are rising among adolescent girls, with the prevalence of severe obesity being four times higher in the most deprived areas versus least deprived areas [7]. If the current trends continue, the World Health Assembly global action plan of a 15% reduction in the prevalence of physical inactivity by 2030 will not be met. This issue is particularly prevalent among adolescent girls in the UK and Ireland, with approximately 90% of girls being insufficiently active [8,9]. Accordingly, it is critical that more PA opportunities become available to meet the needs of adolescent girls, in order to attract and retain their participation during their development through adolescence into adulthood.

There is an extensive body of research exploring the factors that influence youth participation in PA [10,11,12]. Results indicate that factors often differ by gender; for example, competition is a common facilitator for boys but often hinders girls’ participation [11], muscle gain is a common motivator for boys’ exercise but is a barrier for girls [13], and boys engage in more team-based sports while girls typically partake in more individual sports [14]. Evidence suggests girls may experience more PA-related psychosocial issues than boys, with girls commonly citing low self-esteem, low perceived competence and poor body image as deterrents to being active [15]). Past research argues that such issues may be borne from sport and physical education (PE) settings that traditionally celebrate stereotypical masculine characteristics [16], and undervalue girls’ contributions, causing girls to feel marginalized [17]. These concerns are heightened for girls from low socioeconomic areas who, in a recent survey of 50,000 UK adolescents, were found to have significantly lower wellbeing scores than girls from affluent areas (an association that was not found in boys) [8]. Further, Irish boys from all-boys schools were found to be over twice as likely to meet the minimum recommended PE minutes per week in comparison to girls from all-girls schools, with girls from deprived areas receiving almost half the PE time of girls from non-deprived areas [18]. Based on these marked gender and societal differences, future interventions need to consider the factors that specifically impact the target population whom they are targeting. 

The socioecological model [19] is a theory-based framework that has been used to better understand and categorize the various multilevel factors (intrapersonal, interpersonal, organizational and environmental) influencing adolescent girls’ PA [20]. Previous research has found that adolescent girls often experience feelings of low body confidence and self-esteem (intrapersonal) [21,22,23]. In addition, peer pressure to conform to gender-appropriate physical activities can lead to dropout of sport and exercise as peers become more influential (interpersonal) [24,25,26]. PE classes are often centered around team-based sports that celebrate stereotypical masculine traits such as speed, strength and competition (organizational level) [3,13,27]. Furthermore, opportunities for girls to be physically active within the curriculum and in extracurricular activities frequently become less available (environmental) [21,28,29]. Although factors that influence adolescents’ engagement with PA are emerging, recent reviews call for more qualitative research to be conducted with specific subgroups of adolescent girls (e.g., inactive or low socioeconomic groups) to truly understand their needs in order to create effective PA interventions [24,30,31].

We established The HERizon Project as a program of research in response to the aforementioned low PA levels of adolescent girls [6] and given the physical, psychological and social consequences of physical inactivity [32]. Specifically, the HERizon project aimed to develop an effective intervention to increase the PA of adolescent girls in the United Kingdom and Ireland, particularly those who are inactive and from low socioeconomic backgrounds. Following the Medical Research Council guidance on the development of complex interventions [33], the first step of intervention design involved qualitative formative research with the target user. It is crucial that interventions reflect the needs, preferences and ideas of future service users [34], yet there is a paucity of qualitative research with adolescent girls who are most in need of intervention (i.e., those who are inactive and from low socioeconomic areas) [35]. Formative research is a critical in the development and implementation of effective behavior change interventions as it allows detailed information to be gathered about the audience the intervention is being designed for [36]. Collecting information on target user behaviors, interests and needs can help to improve recruitment and retention rates, as well as ensuring the intervention is culturally and geographically appropriate [37]. Formative work has been utilized in the development of many behavior change interventions, including those based in the school [38] and community setting [39]. Therefore, this qualitative formative research study aimed to explore socioecological influences [20] on physical activity behaviors among adolescent girls in the UK and Ireland. These findings will be used to inform the development of The HERizon Project physical activity intervention targeting inactive girls from the United Kingdom and Ireland. 

## 2. Materials and Methods

### 2.1. Research Design

Qualitative focus groups were used to explore the factors that influence adolescent girls’ PA. There is a need for researchers to move away from seeing themselves as the experts and to listen to the target audience in order to identify and respond to their needs [40]. Focus groups gave participants an opportunity to share and compare their experiences of PA, allowed the research team to gather information on girls’ collective views and has been used previously for exploring the determinants of girls’ PA [41]. Reporting of this study was guided by the consolidated criteria for reporting qualitative research (COREQ, [42]) (Appendix A in Appendix A). Ethical approval was granted by the University Ethics Committee (reference: 19/SPS/023), and informed written consent and assent was obtained from parents and participants prior to participation. 

### 2.2. Recruitment

Girls (13 to 17 years) were recruited from Government-funded, non-fee-paying secondary schools and youth clubs in a large metropolitan city with a high deprivation rate in Northwest England and from both rural and metropolitan areas of Ireland. Six schools and six youth club gatekeepers were invited to take part in the study. The final sample included five schools (*N* = 2 NW England, *N* = 3 Ireland), and two youth clubs (*N* = 2 NW England). The participants did not receive an incentive for taking part in the study. 

Information packs containing a participant information sheet, consent/assent forms and a 1-item screening questionnaire were distributed to potential participants. Participant information sheets stated that the study aimed to recruit inactive girls, and girls were able to self-assess their eligibility by ticking whichever of two statements on the screening questionnaire they felt best described their PA: “I am often active and enjoy sports/exercise” or “I am mostly inactive—if I can avoid sports/exercise I will!”. This approach was taken as a means of encouraging participation from inactive girls, while not excluding girls who wanted to take part despite perceiving themselves as active. 

### 2.3. Data Collection

#### 2.3.1. Demographic Questionnaire

Prior to the focus group, participants completed a demographic questionnaire which captured information including date of birth, height, weight, ethnicity and the first three digits of their home postcode for deprivation level [43,44]. 

#### 2.3.2. Focus Groups

Focus groups were used to capture data on the factors that influence adolescent girls’ participation in PA. All focus groups were conducted by the first author (female Ph.D. candidate holding an MSc, trained and experienced in running focus groups), in secondary schools and youth clubs during opening hours. Participants had no prior relationship with the interviewer. Focus groups were predominantly conducted in a classroom where participants could be seen but not overheard [45]. A youth club coordinator sat in the room during one focus group, while all others were conducted with only the researcher and participants. A semi-structured interview guide was developed by the research team and informed by previous literature [28]. The questions were piloted with three adolescent girls to ensure that questions were understood and deemed appropriate for this age group. At the beginning of the focus group, participants were informed that “physical activity” included all sport, exercise, physical education (PE) and any other planned or non-planned bodily movement that elevated their heart rate. Questions focused on what participants enjoyed and did not enjoy about PA, what factors could potentially increase their participation in PA and participants’ views of gender differences within PA (interview guide available in Appendix A). The questions asked during the focus groups and the PA definition used were purposely kept broad to capture what types of PA and settings were meaningful from the perspective of the participants. 

An ice-breaker activity was used to help participants to feel more comfortable speaking aloud within the group [39]. Throughout the focus group, participants were encouraged to share information to the level at which they were comfortable and reminded there were no right or wrong answers and that it was okay if their views were different to those of other participants [46]. In order to maximize participation and group interaction, the researcher attempted to engage all participants by active listening, eye contact and paraphrasing to check for understanding [47]. 

### 2.4. Data Analysis

All focus groups were audio-recorded and transcribed verbatim. The transcripts were imported into NVivo 12.0 and analyzed thematically using an inductive approach in order to encapsulate participants’ shared views of PA and get an overall understanding of their PA experiences [48]. The analysis was conducted by the first author who became familiar with the data by reading and re-reading transcripts. Quotes that were considered to represent a similar meaning or pattern were clustered together into potential themes and subthemes. The second author acted as a “critical friend” by independently reviewing a subsample of transcripts and offering alternative interpretations of the data, encouraging reflection and challenging the initial thematic structure [49]. During this process, we recognized that themes were broadly reflective of the socioecological model [20] which has been used in past research to illustrate the influence different factors have on girls’ participation in PA, i.e., intrapersonal, interpersonal and organizational factors [25]. Themes were mapped to the socioecological model as appropriate. Throughout the coding process, regular meetings took place between E.C., P.M.W., L.F., A.J.M.W. to review, debate and refine themes. 

## 3. Results

### 3.1. Participant Demographics and Group Characteristics

Forty-eight girls returned consent forms, completed demographic and PA questionnaires and participated in focus groups (*N* = 26 Ireland, *N* = 22 England). As shown in Table 1, participants were female, aged between 13 and 17 years (mean 14.8, SD 1.29), and the majority were of white ethnicity (*N* = 42 white, *N* = 4 mixed, *N* = 2 Asian). According to the Pobal HP Deprivation Index [44], 81% of participants from Ireland lived in the most deprived tertile of Ireland. In total, 45% of English participants lived in the most deprived tertile, while 23% lived in the least deprived tertile of the UK according to the Index of Multiple Deprivations [43]. The majority of participants perceived themselves as inactive (71%). 

Eight focus groups (*N* = 4 England, *N* = 4 Ireland) were conducted, ranging in size between 2 and 11 participants (see Table 1 for breakdown of characteristics of each focus group). Focus groups lasted 32 min on average (SD 7.4 min). There were no noticeable differences found between focus groups regardless of group size or location, nor were differences found between participants who perceived themselves as active or inactive.

### 3.2. Factors Influencing PA 

Key themes were identified at the intrapersonal (fear of judgement and changing priorities), interpersonal (changing social pressures and support from others) and organizational level (delivery of PE). In addition, one factor was identified as a cross-level theme (gender inequality). Subthemes are identified below in italics. A visual overview of how the main themes map onto the socioecological model can be seen in Figure 1. Table 2 provides an overview of themes and subthemes with illustrative quotes. 

#### 3.2.1. Intrapersonal

**Fear of judgement.** One of the strongest factors that arose throughout all focus groups was participants’ *lack of confidence in their PA skills,* which caused them to avoid attempting new activities or stop PA completely due to a fear of criticism. Participants explained that they would feel “ashamed” if they were to exercise in public and that they would be more comfortable exercising in the privacy of their homes. Many girls explained that they did not like to get involved in team activities, regardless of their sporting ability, as they felt too much pressure to perform well and that they were being *compared to others*. Older girls expressed interest in returning to sports in which they were involved at a younger age, but they were concerned that they would not be able to keep up with others who have more experience. Most participants stated that they would feel anxious *being alone* in a new PA environment and would be more inclined to go if they had a friend. Some girls stated that even if they were a regular member of a sporting club, they would be more likely to also miss the session if close friends were not attending training. Many girls said that they would only try new PA opportunities with close friends whom they trusted to not judge their abilities or bodies.

Each focus group raised issues of *body image* and explained that these insecurities often prevented them from taking part in PA. Physical insecurities included weight, height and shoe size. Many explained that they felt intimidated and uncomfortable attending PE classes and local gyms. The girls felt a *pressure to look good* when exercising, even when they were with fellow female peers. The issue of sweating, being flushed after vigorous exercise and the discomfort of wearing ill-fitting PE uniforms were highlighted as a serious barrier to participation, with some girls explaining that they would not engage in vigorous levels of PA for fear of sweating in front of others. 

**Changing priorities.** Girls in all focus groups spoke of having a lack of free time since entering secondary school and recognized their priorities had changed since becoming a teenager. Most girls explained they were actively involved in sports when they were younger but since reaching adolescence, they had dropped out so they could spend their free time with friends. There was a general acceptance among girls in all focus groups that *academic pressure* increased as the girls progressed into secondary education, especially in more senior year groups. Many girls felt they received conflicting messages from teachers as they encouraged students to stay physically active but also gave so much homework that there was insufficient time for girls to attend classes or team training. Further, there was a general lack of motivation toward PA, with many girls branding themselves “lazy”. 

#### 3.2.2. Interpersonal

**Changing social pressures.** Some girls cited *social influence* to be the cause of dropping out of sports as their friends were no longer participating. Many girls spoke about being afraid of being excluded by peers and missing out on social events due to sporting commitments. Girls said that over time, they eventually prioritized spending time with friends and gave up their sports. 

**Support from others.** Teachers who encourage adolescents to be active and who provide a source of *accountability* were identified as facilitators to maintaining PA. *Peers* had the potential to add to PA enjoyment, as well as diminish it. Participants felt most comfortable around peers that were of a similar skill level and in the company of people who could be trusted not to judge their abilities or appearance. Peers who were not within the participants’ close friend circle could often have the opposite effect and discouraged girls from being active, as they felt they were being judged and at risk of being ridiculed. Many participants discussed anxiety during *PE classes*, particularly in mixed gender classes as they did not feel comfortable exercising in front of boys. Girls also felt anxious when separated into groups without their friends and said that they tended to not fully engage in activities as they were afraid of criticism and judgement from others. 

#### 3.2.3. Organizational

**Delivery of PE.** Adolescent girls in all focus groups identified that *lack of autonomy* in PE, including not having a say in what type of PA they engage in, when they do it and who they participate with, was as a key barrier to being physically active. Most participants expressed a strong resentment toward being forced to do activities, and when given the choice, they felt more respected and more inclined to engage. All focus groups said they felt PE was *not delivered in a “fun” way*, that it was repetitive and boring. The factors that make PE enjoyable were consistent across all focus groups and included activities being varied and informal. The *timing* of PE lessons was also seen as a barrier to girls’ participation. Most said that they were not given enough time after PE to shower and get changed. Further, if PE was timetabled in the morning/ middle of the school day, the girls were unlikely to engage as they did not want to sit in other classes afterwards, as they felt uncomfortable without showering. Two girls spoke of PE having little *priority within the school timetable;* PE was often cancelled during exam times, and less PE scheduled for senior year groups. Girls within one focus group identified the school’s *poor facilities* as a major barrier to their participation in extracurricular activities.

#### 3.2.4. Multilevel (Crosses Interpersonal, Organizational/Environmental and Policy/Cultural)

**Gender inequalities.** Girls in mixed schools felt that *boys actively excluded* them from PE based on their gender. Due to this, girls felt that they were incompetent when it came to being physically active because they are female. When girls were included, the boys were perceived to be very rough, and a couple of girls reported being injured during mixed-gender activities. Participants also highlighted their belief that schools and teachers offered *less support* to girls, and that girls are not encouraged to be active, nor are they celebrated for their sporting achievements in comparison to boys. Similarly, all girls highlighted the *lack of PA opportunities for girls*. Some of the older girls reported that there were no senior teams for women for the sports they played. They felt that this lack of opportunity forced them out of the sport. In seven out of eight focus groups, there was a general consensus that *girls “sit out”* and do not participate in PE. Although this was not passively accepted by teachers, it was reported that some girls had never participated in PE and there was no repercussion for non-attendance.

A number of participants felt they would participate more and would feel more comfortable having *female-only* PA facilities, PE lessons and activities. Participants in one focus group also brought up the issue of religion and how this might impact their participation in PA if there were no single-sex PA opportunities. Participants explained that there are few female role models who have *professional sporting careers.* Most focus groups, regardless of participant age, identified how the gender pay gap (e.g., professional female athletes earning significantly less than males) caused them to feel unmotivated should they aspire to become a professional athlete. Participants felt there is a *social stereotype* of what is expected of girls in comparison to boys (e.g., spending free time playing with friends or going to team training is acceptable for boys, but for girls, they felt they were expected to prioritize studying in their spare time over being physically active). Most participants said they spent their leisure time with friends “hanging around”. When asked if they were active when out with their friends, they said it was not acceptable in their local areas for girls to be seen on bicycles or with footballs. 

## 4. Discussion

This study adds further insight into the factors that influence girls’ participation in PA; however, it is notable that these factors are reflective of those some two decades ago [11,50,51,52]. As far back as 1998, researchers were calling for the PE setting to be rejuvenated and the curriculum reformed to address the gender gaps [53], but based on current results, it appears this gap has merely been managed, not reduced. This is a significant issue as the negative experiences that girls have in PA, sport and PE can have a lasting effect on their engagement with PA across the lifespan [17]. In accordance with past studies, social support and autonomy were key facilitators to girls’ PA [54]. Numerous barriers were identified by participants at all levels of the socioecological model, including intrapersonal factors (e.g., the fear of being judged), interpersonal factors (e.g., negative experiences of PE) and multilevel factors (e.g., societal gender norms), which are consistent with past literature [13]. These findings will be used to inform the development of a future intervention (The HERizon Project) which is aimed at increasing girls’ PA participation. 

### 4.1. Intrapersonal Factors

In a recent UK survey with 21.000 girls, one third said they did not take part in PA due to low self-confidence in their physical abilities, and a further third avoided PA because they felt their bodies were being scrutinized by others [55]. These intrapersonal issues were reflected in the current study as a fear of judgement was an overriding theme across all focus groups. Although being skilled can make PA participation easier [16], girls may still not engage due to fear of comparison with others [11] or having low perceived competence [56]. If girls feel insecure about their abilities, they are unlikely to participate in order to preserve an image of competency and to avoid any potential embarrassment [30]. By not taking opportunities to develop skills, they further diminish their confidence and compound their fear of being judged by others [57], which can impact their future PA [58]. This was seen in the current study as girls said that they would like to join sports clubs but felt they did not have the required skill level nor want to practice in front of others that are more experienced.

In a study with 524 girls from low socioeconomic areas, 72% were found to be dissatisfied with their body image, regardless of being in a “normal” BMI weight category [59]. Negative body image is a prominent barrier to girls’ PA [20] and has been found to start in girls as young as 7 years old [55]. Girls in the current study felt that PE was a costly risk that leaves their physical appearance open to jeering by peers in their class, especially boys, and therefore, they often chose to not participate. Boys’ perceptions of girls being active may exacerbate girls’ body concerns as in one qualitative study, boys admitted to calling girls “disgusting” and “nasty” if they sweat while exercising [60] (p. 87), and another work found that boys will intimidate girls who are overweight by teasing and excluding them [25]. Boys were found to exclude girls who rivaled their strength, with one boy explaining she was “too big and too tall… she made most of the goals and made the girls beat the boys” [40] (p. 42). In this same study, another girl was excluded from playing by boys because she did not know how to run fast [40]. Teachers in a recent study were also found to hold stereotypical gendered views of bodies as teachers described a fictional female student who wanted to take part in football over dance as someone with short hair, a tomboy and stronger and heavier than other girls [61]. In order to support the development of a healthy body image, it is important for professionals, such as PE teachers and community coordinators, to reflect on their own bodily gender biases, to educate girls on body appreciation and to support boys in increasing their knowledge and understanding of bodily related concerns and the relevance of their own behaviors within a PA context. 

### 4.2. Interpersonal Factors

Interpersonal interaction is a commonly cited facilitator to girls’ PA [20], and although being active with friends can enhance enjoyment, and thus increase participation [54], this study found the relationship is nuanced and complex. Girls in the current study explained that only specific close friends, whom they trust and feel comfortable around, have positive effects on their PA and that other friends and classmates can have the opposite effect, causing girls to be deterred from team-based activities and many preferring to exercise at home. This finding corresponds with past studies which indicate girls disengage from PA due to feeling anxious when asked to form groups in PE lessons [56], as a result of feeling pressure to not let teammates down [16] and because of general exclusion from other girls [62]. The peer contagion model illustrates how adolescents often mimic the health behaviors of their peers [63]. Although in some scenarios, this can be beneficial (i.e., active girls often have active friends [64]), many girls in the current study said that they had dropped out of sports because their friends no longer did it, a trend well documented in past literature [65]. Girls feel under pressure to conform with what is viewed as gender-appropriate by friends, and usually, this does not involve engaging in sport [61]. Girls that push against these stereotyped norms are at risk of being excluded and victimized by their peer group [66]. Given the complexity of the influence social interaction has on girls’ PA, it has been recommended that future PA interventions examine the factors that mediate the relationship between social support and PA participation [54]. 

#### 4.2.1. Organizational/Multilevel Factors 

Girls’ PA behaviors are regulated by organizational factors including societal norms that scrutinize what activities girls should and should not engage in [67,68,69]. It has been argued that it is not the sport or activities themselves that are the issue but instead how these environments are constructed that leads to girls’ disengagement [70] as PE and sport settings typically celebrate stereotypical masculine traits [71] with boys’ achievements and activities given higher status [72]. Girls in the current study felt their PA participation was limited by the opportunities available to them and by the gender expectations of society. Researchers note that as society often portrays femininity as being incompatible with sport, many girls drop out [73], and those that do not conform to these gender stereotypes often face exclusion and victimization by peers [74]. Although efforts have been made to bridge the PA gender gap (Sport England This Girl Can campaign and the Federation of Irish Sport 20 × 20 campaign), a divide still remains as girls are often marginalized and undervalued [75], and boys are perceived to have access to better facilities and support [10,76,77]. Girls were pessimistic about the likelihood of becoming professional athletes. Marginalization of females from recreational PA and elite sport contributes to female professional athletes being seen as outlying trailblazers rather than a normal occurrence [78]. As girls transition to secondary school, there are fewer opportunities for girls to engage in traditionally masculine activities, such as football and rugby [79]. Even when such opportunities become available, many girls choose not to participate in order to avoid negative comments from peers [80]. Consequently, girls in the current study said they would feel more comfortable in female-only activities. This echoes past work which stated that there has been “nothing more constraining and alienating” for girls than a coeducation and multiactivity PE curriculum [69] (p. 32). Women in mixed-gender gyms often receive unsolicited advice and attention from males, which leads to feelings of discomfortable, and in some situations prevents them from exercising [26]. To support girls, it has been recommended that they be given opportunities to engage in a range of flexible PA opportunities in a separate female-only space [81]. This is contested by others who argue that gender segregation in PE is further reinforcing gender stereotypes and instead suggest creating a “homely” space that is social, intimate and emphasizes acceptance [81] (p. 359). An environment that aims to normalize differences [82] and provides opportunities for boys and girls to have positive PA experiences together can help to make either group more aware of each other’s capabilities [74]. 

#### 4.2.2. Implications for Intervention Development

This study identifies some the multilevel factors that influence adolescent girls’ PA and can suggest practical recommendations to inform the development of future PA interventions, including the HERizon Project. In order to create effective and meaningful PA programs, adolescent girls’ voices should be at the core of their development so that their needs and interests at each socioecological level are catered for. 

The findings support the necessity to provide PA opportunities that focus on intrapersonal development, including girls’ perceived competence and reducing feelings of judgement from others. Providing girls with the autonomy to choose from a range of different fun and diverse activities, as well as giving them the independence to choose when and where they will be active, is important for increased participation. Fear of comparison can be reduced by providing PA opportunities that focus on self-progression through appropriate and attainable challenges and that are held in a safe, informal environment. As there is a lack of home-based interventions, further exploration of this setting may help to overcome some of the aforementioned barriers, as well as body-image-related concerns, such as avoiding vigorous exercise for fear of sweating in front of others. 

Positive interpersonal experiences, such as socializing in an inclusive and diverse environment, may be more appealing to girls than competitive sports settings. Many girls spoke of being fearful of joining PA opportunities alone, and remote interventions may provide the space for girls to have the virtual support of friends yet remain shielded from judgement of their bodies and skills. Mentors were identified as being facilitators who could provide encouragement and accountability for girls to develop and maintain PA habits. Mentors who cultivate a culture of acceptance and demonstrate an awareness of the pressures girls feel under to conform to societal gender norms may help girls to feel supported and nurture their PA participation. Segregating by gender in PE and other community sports may reinforce gender stereotypes. Instead, it is suggested that girls and boys be offered equal PA opportunities in school and community settings, while also celebrating girls’ sporting achievements and promoting female role models. Further, where possible, girls may feel more comfortable being instructed by females, especially in subgroups where religion may prevent them from taking part in certain activities. 

Although the focus of this study was physical activity, many of the girls primarily spoke of school PE and their negative experiences. We must recognize the hard work many PE teachers are doing to create equal access and opportunities for young women and highlight the need for upper management, national curriculum developers and governing policies to support them in this ongoing battle. Within schools, PE should be prioritized in the timetable so that there is sufficient time to engage in activities and ample time after class to change back into school uniform. To overcome some of the cited issues, it is suggested that there be a more open dialogue within the PE setting as often the perception girls have of the opportunities available to them is markedly different to the perceptions of teachers [83]. Our past experiences lead to our own conscious and unconscious biases, and so, it is recommended that PE teachers reflect on their own prejudices that may be reinforcing the gender divide in the PE environment. By actively encouraging girls’ participation in class and breaking away from traditional gender-appropriate activities, teachers can express their awareness of societal gender inequalities and support girls in challenging social expectations. Future interventions, including the HERizon Project, should offer a range of activities that include traditional “masculine” activities such as strength and conditioning activities, as well as provide support and awareness of the gender barriers girls often feel constrained by. 

#### 4.2.3. Strengths and Limitations

We were successful in recruiting girls from a number of geographical locations across the UK and Ireland. This helps in gaining a better understanding of PA determinants for adolescent girls and increases the generalizability of the study’s findings. However, it must be acknowledged that the majority of girls were of white ethnicity, and no participants had physical disabilities. Therefore, future work should include a more diverse sample of participants to reflect the diversity of Irish and British cultures. Further, although the study aimed to recruit inactive girls from low socioeconomic areas, approximately a quarter of participants were considered active and were from affluent areas of the UK and Ireland. This may be due to issues with the recruitment strategy, including difficulties in obtaining returned consent forms and screening processes. Due to the purposeful attempt to recruit inactive girls, most active girls self-excluded from taking part in the study, and therefore, the findings may not generalize to more active groups of adolescent girls. Although the target sample size was reached, the number of participants in each focus groups ranged, with one group only containing two participants. Although the findings from this group were consistent with those of larger groups, it is acknowledged that discussion may have been limited due to the small number of participants. Participants predominantly spoke of PA in relation to school PE. This may have been due to many of the girls viewing themselves as inactive, and therefore, their main experience of PA was within the school PE setting. Another explanation is likely due to the majority of focus groups being conducted within the school building during class time. Future research should consider if another setting is more appropriate to capture girls’ experiences of non-school-related PA, but researchers should be mindful of the pragmatic difficulties of recruitment in community settings, such as youth clubs, due to their unstructured nature, i.e., attendance is not compulsory, which may lead to organization issues. Although this research aimed to explore girls’ PA experiences, it may have been beneficial to also speak with others involved in the provision of girls’ PA, such as PE teachers and youth club coordinators. Given that adolescent girls’ PA behavior is complex, it is important to have a broad understanding of the issue, and speaking with other stakeholders may help to develop a more holistic view of the issue.

## 5. Conclusions

The current formative research provides an insight into the multilevel factors that influence adolescent girls’ participation in PA. These findings will be used to inform the development of the HERizon Project, an intervention targeting adolescent girls’ physical inactivity. Adolescent girls experience numerous challenges that often deter their sustained participation in PA. Girls’ PA levels were influenced by a fear of being judged, changing priorities and social pressures, support from others, the delivery of PE and gender inequality. Many of these factors have a negative influence of girls’ PA and stem from gender-biased societal values. In order to increase PA participation for adolescent girls, interventions that are holistic and consider factors on each of the socioecological levels are needed. Interventions should be set in a location where girls feel comfortable, with a focus on enhancing wellbeing and enjoyment rather than competition and comparison with others. 

## Figures and Tables

**Figure 1 children-08-00031-f001:**
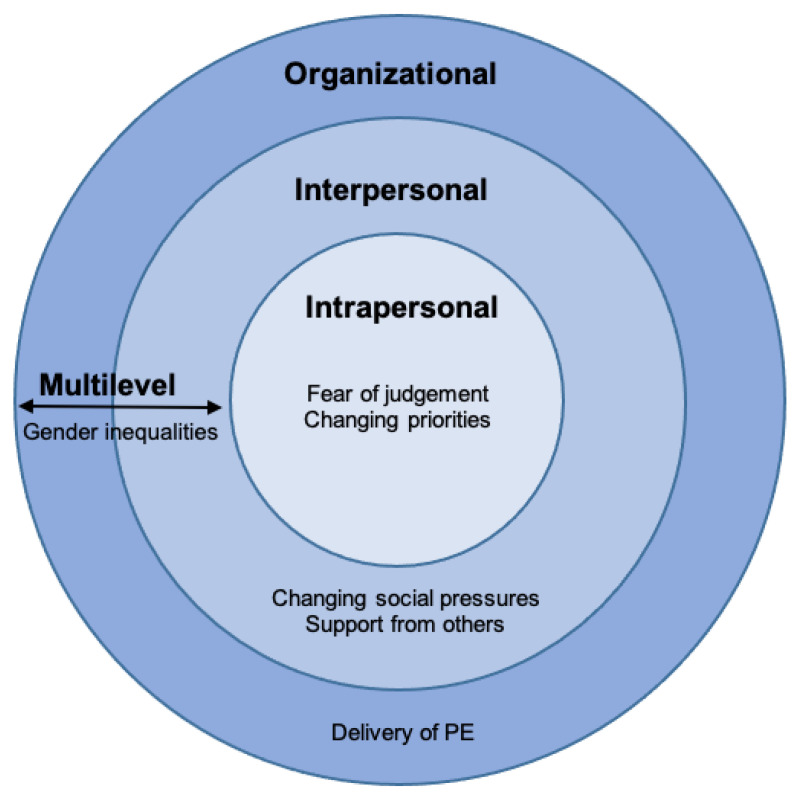
Adapted socioecological model of health behavior change (Sallis, 2005), including example factors of physical activity at each level. The model suggests that the health behavior being targeted should be specific and that interventions should be designed using multiple levels.

**Table 1 children-08-00031-t001:** Focus group participant demographics.

Group	*N*	Country	Setting	Type of School	Age (Years)	Perceived PA Status *
Total	48				13–17	34 inactive/14 active
1	4	England	Youth club	Mixed sex	13–15	2 inactive/2 active
2	4	England	Youth club	Mixed sex	13–16	2 inactive/2 active
3	9	Ireland	School	Mixed sex	14–17	7 inactive/2 active
4	10	Ireland	School	Mixed sex	13–15	6 inactive/4 active
5	5	Ireland	School	Girls Only	16–17	5 inactive
6	2	Ireland	School	Girls Only	17	2 active
7	7	England	School	Mixed sex	14–15	6 inactive/1 active
8	7	England	School	Mixed sex	15–16	6 inactive/1 active

* According to 1-item self-report screening questionnaire.

**Table 2 children-08-00031-t002:** Factors influencing adolescent girls’ participation in physical activity (PA).

Socioecological Model Level	Theme and Subtheme	Demonstrating Quote
Intrapersonal	**Fear of Judgement**	
	*Lack of confidence in skills*	Even if everyone thought you were good at [sport] you wouldn’t think you were good at it … you could feel like people are laughing at you (P15, FG3, perceived inactive).
	*Comparison with others*	You want to see [others] are skinnier than you… Others are better… you might feel like you are too old compared to the others (P32, FG5, perceived inactive).
	*Being alone*	People feel less insecure when they are with their friends, they feel more insecure if they are alone and people are staring at them (P5, FG2, perceived active).
	*Pressure to look good*	In PE you get loads of pressure when you do it in school… you always feel under pressure because you have some of the skilled people watching you [doing PE] and you need to be your best at everything and you can’t screw up (P27, FG4, perceived inactive).
	*Body Image*	I know girls who dropped out of sport because they don’t want to look all muscly… Imagine going home with a bright red face, sweat rolling off you and everything (P43, FG8, perceived inactive ).
	**Changing priorities**	
	*Make-up instead of sport*	It feels like girls are not meant to do exercise… rather exercise with their fingers doing their makeup (P3, FG1, perceived inactive).
	*Academic pressure*	They say during the [state exams] to “keep up your sports, keep up your sports” but then when you come into school they are like “study, study”, (P34, FG6, perceived active).
Interpersonal	**Changing social pressures**	
	*Social influence*	[PA] kind of changes by your age, you want to do whatever your friends are doing so you stop whatever [PA] you were doing (P20, FG4, perceived inactive).
	**Support from others**	
	*Accountability*	Researcher: What would help you to be more active? If you feel like you are getting support and you’re getting pushed and someone is motivating you to do it. A group chat with all your mates [would be helpful] so you could say like “I’m going to the gym”, like help me on this so I can get better at it (P36, FG7, perceived inactive).
	*Peer support*	On your own I don’t think you would be doing [PA] as much as if you were with your friends, you’d be more motivated with your friends because they are doing it too (P3, FG1, perceived inactive).
Organizational	**Delivery of PE**	
	*Lack of autonomy*	Some people just aren’t into running and they are getting forced to go out and do that when they would rather be in school doing team sports or something. I just think it shouldn’t be compulsory to do certain things (P34, FG6, perceived active).
	*Not delivered in a “fun” way*	It’s the same stuff all the time… they say we can’t be bothered but we just want to do something active and fun instead of doing rubbish stuff (P45, FG8, perceived active)
	*Timing*	We only get an hour to do sports… we have to be changed, put up the nets and we only have half an hour because we still have to do a warm-up… on top of that you have homework and getting your books… so it’s kind of pointless to keep [PA] up (P29, FG5, perceived inactive).
	*Priority within timetable*	[We] should have more physical activity a week… most lads teams will get more training a week whereas in here it’s like once a week (P5, FG2, perceived active).
	*Poor facilities*	We don’t have really good facilities… our hall is half the size it’s meant to be so it’s hard to do actual sports because when you go to matches its completely different (P29, FG5, perceived inactive).
Multilevel	**Gender inequalities**	
	*Exclusion by boys*	[The boys] never pass you and the girls are just like in the way for them (P23, FG4, perceived active).
	*Less support for girls*	If the PE teacher’s like “pick what game you want to play“ and the girls say hurling and the guys say soccer they are going to go with the guys… they don’t listen to what the girls have to say (P26, FG4, perceived inactive).
	*Lack of PA opportunities for girls*	[Schools] don’t really have girls’ stuff, they mostly have things to do with lads, they don’t encourage the girls to go and do football, they are doing it for the boys (P4, FG1, perceived active).
	*Girls ‘sit out’*	Researcher: Do most girls in your class take part in PE? No, no [girls] do [PE]. In our class there are 14 girls and like 5 of us that do [PE]. We all just sit in [the hall] at the sides and just refuse to do it (P45, FG8, perceived active).
	*Need for female-only PA opportunities*	Some people don’t like to be with other gender and most facilities aren’t enclosed (P5, FG2, perceived active).
	*Professional career*	[Girls doing sport] is frowned upon, even on TV, girls are paid less, and boys are paid more, there isn’t enough media coverage (P6, FG2, perceived active).
	*Social stereotypes*	Boys can ride a bike when they’re out, imagine one of us going around on a bike? Are you mad?... If a girl got seen driving a bike around here you would probably get robbed… it’s not normal [to ride a bike if you’re a girl]… if you were seen riding a bike at the age of 12 they’d come over and knock you off (P45, FG8, perceived active).

Quotes are assigned to participants on the basis of which focus group they were in and whether the girl perceived herself to be active or inactive (e.g., Participant 3 (P3), Focus Group 1 (FG1), inactive).

## Data Availability

Data presented in this study are available on request from the corresponding author. The data are not publicly available due to privacy.

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
