# Peer review of "“Girls Aren’t Meant to Exercise”: Perceived Influences on Physical Activity among Adolescent Girls—The HERizon Project"

_children, 2021, doi:10.3390/children8010031_

Round 1

Reviewer 1 Report

This is a well-written manuscript that addresses the important issue of adolescent girls’ PA, that up to now were unable to successfully cope with. This study adds to the understanding of perceptions of girls on PA and might be helpful for intervention developers to change their thoughts on how to deal with this challenging problem. Yet, I do have a couple of questions and comments, which I’ll point out one-by-one below.

Comments

Line 97: Replace girl’s for girls’

Line 136–138: The authors refer to a 5-pouint Likert scale, yet their data categories indicate a range of 6 (0= least active; 5= most active). Please correct.

Line 146-147: Could the authors reflect on the reliability of the focus groups? Technically, it seems more appropriate to conduct focus groups with an interviewer and secretary, that supports the interviewer during the process. Is it possible that due to specific interests in the HERizon Project results are slightly biased?

Line 150-152: The broad definition of PA was used in the focus groups. Did this influenced the content discussed in each of the focus groups as this broad approach may lead to more superficial answers on specific settings or types of PA. Were the different types of PA and settings in which PA mostly takes place discussed separately, e.g. PE? This also reflects to lines 400-402 in the discussion section in which the authors point out that PA is too often seen as team-based sports. I feel the use of the definition of PA in the focus groups needs to be addressed more clearly.

What was the duration of the focus groups? Did the participant received an incentive?

How were the focus groups composed? Were there any criteria taken into account to create homogeneous groups in which the girls felt save to fully express their thoughts about this topic (rather than remembering them that there were no good or bad answers)? Particularly as girls indicate themselves that they experience fear of judgement, it may be important to consider homogeneity in groups. It looks like some groups contained of an equal number of girls that perceived themselves as active and as inactive, while some groups mostly (or even only) consisted of girls considering themselves as inactive. Did outcomes between these groups differ?

Line 196 Table 1: What about focus group 6? How reliable are outcomes from a focus group containing 2 participants? Moreover, there seems to be a typo in group 5. I guess 1/6 should be 1.6.

Table 2: The statement ‘active or inactive’ is lacking in the quote about body image

Based on the results stated in lines 208-210, it feels that a reflection on the impact of this avoidance would be appropriate in the perspective of a negative vicious circle girls seem to be in. Avoidance prevents experiences and increases inactivity, negatively influencing their motor competence, likely leading to lower confidence in skills, leading to even more avoidance and so on. This further highlights the necessity to act. In this perspective prevention of these girls' negative feelings towards PA could be acknowledged more as is it is likely that these feelings may reveal during early adolescence.  

Reviewer 2 Report

I would like to thank the authors for engaging in timely and important work. This was an interesting and very well-written paper exploring girls' perceptions of PA in the UK and Ireland. Overall, I believe the study has merit and offers insight for those seeking to create programming for adolescent girls to be active. I do have some concerns about the study, which I outline below. 

  • My primary concern is related to the overall contribution of the manuscript to the literature. Many of these dynamics are already well established in the literature. Thus, I would encourage the authors to more clearly articulate the advancement provided by the manuscript. 
  • Also, the authors note the use of the Social-Ecological Model as a frame for the study. However, the findings are not meaningfully discussed in this framework. These findings should be situated in the model- perhaps with a more explicit discussion of the nature of the findings spanning across the many layers of the model. Perhaps the authors may also consider a figure, in which the themes are appropriately mapped onto the concentric circles or some other graphic depiction of the SEM as an organizing structure of the findings? 
  • The authors describe data collection of quantitative data and (at least tangentially) note the non-significance of difference between 'active' and 'non-active' participants. However, no analytic technique or information is provided for these analyses. They should be omitted or fully explained. That is, the authors note there were no differences between them, without fully explaining the means by which the conclusion was reached. 
  • Also related to methods, the authors indicate an interesting recruitment method, whereby they explicitly recruited non-active participants but accepted both self-identified active and non-active individuals. How might this limit the findings of the study, as many 'active' girls may have self-selected out of the study? Do the authors have a sense of how many participants could have been lost? Or, if they were somehow all retained, this needs to be more clearly articulated in text. 
  • Finally, why is the HERizon project mentioned? The only thing ever said about it is that this study will inform the project. While a worth use of the data- it is not relevant to the study itself. 

Overall, I enjoyed this paper. Again, I thank the authors for their engagement in this interesting and important topic. 

Round 2

Reviewer 2 Report

Thanks to the authors for their thoughtful revisions to this manuscript. I believe these changes have made an overall much stronger manuscript, that is well situated in the literature, while also articulating its purpose. Further, the elimination of the more methodologically problematic elements (i.e., the quantitative elements) has made for a cleaner overall paper. I have no remaining major concerns with the manuscript as is presented. 

One note, however, in the authors' response document they note the 'purpose of qualitative research is not to produce reliable or valid data,' in response to the other reviewer's note on having a secretary present. While I agree that focus groups can be conducted with one individual researcher (and thus, see no issue in the paper), I would argue that qualitative research is concerned with similar concepts, often termed 'trustworthiness and credibility.' For example, your use of the critical friend approach is indeed an attempt to ensure these things- as it (in part) helps reduce potentially biasing factors related to misunderstanding the data itself.